# Disseminated Infection with *Aspergillus fumigatus* in a Scarlet Macaw Parrot (*Ara macao*)—A Case Report

**DOI:** 10.3390/ani14152282

**Published:** 2024-08-05

**Authors:** Oana Irina Tanase, Geta Pavel, Ozana Maria Hritcu, Mihaela Anca Dascalu, Bianca Elena Bratuleanu, Cristina Mihaela Rimbu, Florentina Daraban Bocaneti

**Affiliations:** 1Department of Public Health, Faculty of Veterinary Medicine, Iasi University of Life Sciences Ion Ionescu de la Brad, 700489 Iasi, Romaniacrimbu@yahoo.com (C.M.R.); 2Department of Preclinics, Faculty of Veterinary Medicine, Iasi University of Life Sciences Ion Ionescu de la Brad, 700489 Iasi, Romania; 3Regional Center of Advanced Research for Emerging Diseases, Zoonoses and Food Safety (ROVETEMERG), Iasi University of Life Sciences Ion Ionescu de la Brad, 700489 Iasi, Romania

**Keywords:** mycosis, *Aspergillus*, macaw

## Abstract

**Simple Summary:**

Aspergillosis is an important fungal disease occurring in avian fauna, especially in birds kept in captivity. In *Psittaciformes*, severe disease occurs in the lungs and air sacs, with the development of white-to-yellow caseous nodules and plaques in the organs, in addition to greenish-grey fungal growth in the air sacs. Herein, we report the presence of disseminated infection with *Aspergillus fumigatus* in a 3-year-old male scarlet macaw parrot (*Ara macao*) that was presented to the Exotic Animal Clinic at the Faculty of Veterinary Medicine, Iași University of Life Sciences (Iași, Romania) for its postmortem examination. The confirmation of the fungal infection was achieved using histopathological, microbiological, and molecular methods. Since birds suffering from *Aspergillus* spp. do not always show respiratory issues, or their clinical signs are non-specific, this may create diagnostic difficulty for clinicians unfamiliar with the parrots’ pathology. Therefore, for a definitive diagnosis, the demonstration of fungal presence by cytology or histopathology and its identification using culturing and molecular techniques is required.

**Abstract:**

A 3-year-old male scarlet macaw parrot (*Ara macao*) was presented to the Exotic Animal Clinic at the Faculty of Veterinary Medicine, Iași University of Life Sciences (Iași, Romania) for its postmortem examination. According to the owner, the parrot had been raised only in captivity and after 5 days of inappetence, lethargy, and mild respiratory clinical signs, the parrot died. The post mortem examination revealed various-sized granulomas and caseous plaques in the lungs, air sacs, spleen, intestinal serosa, and liver. Microscopically, the granulomas were characterized by a necrotic center and the infiltration of numerous multinucleated giant cells and epithelioid-like cells and by the presence of hyphae typical of *Aspergillus* spp. Moreover, in the liver tissue, a diffuse inflammation, with numerous fungal hyphae, was noted. The fungal culture and the PCR assay allowed for the isolation and identification of *Aspergillus fumigatus* from the lung and liver samples. The macroscopical lesions and the histopathological findings, with the fungal isolation and molecular confirmation of *Aspergillus fumigatus* by nested PCR, provided the basis for the diagnosis of disseminated aspergillosis. To the authors’ best knowledge, this is the first report of disseminated infection caused by *Aspergillus fumigatus* in a scarlet macaw parrot (*Ara macao*).

## 1. Introduction

Aspergillosis is an important fungal disease in avian fauna. Out of the approximately 340 accepted *Aspergillus species* (*Aspergillus* spp.), only a small number are implicated in the development of avian aspergillosis [1]. According to recent studies, *Aspergillus fumigatus* is by far the most prevalent species, representing up to 95% of all occurrences in both wild and domestic avian species [1,2]. Birds kept in captivity are more susceptible to acquiring an infection with *Aspergillus* spp. due to several factors related to environmental conditions (increased concentration of fungal spores precipitated by humid, warm, dirty, poorly ventilated air) or to host immunity (concurrent infections, virulence of the isolate, therapies, stress) [3,4]. The anatomy and physiology of the avian respiratory system is substantially distinct from that of the bronchoalveolar lungs of mammals. Given the small size (2–3 µm), inhaled *Aspergillus fumigatus* conidia is able to overcome the first physical barriers and profoundly penetrate into the respiratory system and subsequently, into the air sacs, which are particularly predisposed to contamination, since they are subjected to an airflow that facilitates particle deposition. In addition, avian species have few resident macrophages to eliminate corpora aliena and have an epithelial surface nearly lacking in a mucociliary transport mechanism [5]. When the immune response is less effective, the fungus can spread from the respiratory system via the circulatory system or by simple extension from the air sac wall to contiguous organs or cavities. Moreover, hematogenous transmission is achieved by hyphal penetration of the lung blood vessels and by means of macrophages transporting viable spores. Under appropriate aerobic conditions, fungal asexual reproduction within the air sacs is a common result, associated with plaques becoming velvety and changing color, depending on the *Aspergillus* spp. involved [6]. Microscopically, lesions are characterized by granulomatous inflammation associated with the fungal hyphae.

Acute infection, usually seen in young poultry, is defined by high morbidity and mortality, by respiratory signs, or by systemic disease in the visceral organs and the brain, while chronic respiratory infection is mostly seen in adult or captive birds, such raptors or parrots [7]. Furthermore, aspergillosis has been confirmed in various species of captive parrots, such in African grey parrots (*Psittacus erithacus*), blue-fronted Amazon parrots (*Amazona aestiva*), or Eclectus parrots (*Eclectus roratus*) [8,9,10]. In *Psittaciformes*, severe disease occurs with infection of the lower respiratory system (lungs and air sacs), resulting in the development of white-to-yellow caseous nodules and plaques in the organs, in addition to greenish-grey fungal growth in the air sacs [11]. Herein, we report the presence of disseminated infection with *Aspergillus fumigatus* in a scarlet macaw confirmed by histopathological, microbiological, and molecular methods.

## 2. Materials and Methods

### 2.1. Case Description

A 3-year-old male scarlet macaw (*Ara macao*) was presented to the Exotic Animal Clinic at the Faculty of Veterinary Medicine, Iași University of Life Sciences (Iași, Romania) in June 2021 for its postmortem examination. According to the owner, the parrot had been raised only in captivity and after 5 days of inappetence, lethargy, and mild respiratory clinical signs, the parrot suddenly died. No antemortem assays had been performed. A post-mortem examination was performed, and various-sized granules were noted in the internal organs. For the histopathological evaluation, tissue samples were collected from the lungs and liver, fixed in 10% buffered formalin, embedded in paraffin, sectioned at 5 μm, and stained with Masson’s trichrome stain. Moreover, in order to visualize the suspected fungal hyphae, the sectioned samples were stained using periodic acid Schiff (PAS) staining. In parallel, for cytological examination, fresh smears from the lungs were prepared and routinely stained using the May–Grünwald–Giemsa method.

### 2.2. Isolation and Identification of Aspergillus *spp.*

Samples from the lungs and liver were cultured onto Sabouraud dextrose agar (Oxoid, Basingstoke, UK) and onto Mueller–Hinton agar with 5% sheep blood (Oxoid) and aerobically incubated at 37 °C for 7 days. *Aspergillus fumigatus* is a thermophilic species that can develop at temperatures of up to 55 °C, but can also survive up to 70 °C [12].

The suspected *Aspergillus fumigatus* isolate was obtained, and its macroscopic and microscopic morphologies, such as hyphae, conidial heads, and arrangements, were identified by lactophenol cotton blue staining, as previously described by Thom and Raper (1945) [13].

The conventional polymerase chain reaction (PCR) technique was performed on lungs and liver for the identification of *Aspergillus fumigatus*, following a protocol published by Sugita et al. (2004) [14]. Briefly, the DNA was extracted using the PureLink™ Genomic DNA Mini Kit, following the manufacture’s instruction. The genus specific amplification was performed using the PCR Master Mix Platinum II Hot-Start Green (Invitrogen, Vilnius, Lithuania), 5 µL of extracted DNA, and 20 µM of primers ASAP1 (5′-CAGCGAGTACATCACCTTGG-3′) and ASAP2 (5′-CCATTGTTGAAAGTTTTAACTGATT-3′). The cycling conditions consisted of: an initial denaturation step at 94 °C for 4 min, 30 cycles of denaturation at 94 °C for 1 min, annealing at 55 °C for 2 min, and extension at 72 °C for 90 s, followed by a final extension step at 72 °C for 10 min. By nested PCR, for the specific amplification of the *A. fumigatus* ITS1 region, the primers ASPU (5′-GCCCGCCGTTTCGAC-3′) and AFI3 (5′-CCGTTGTTGAAAGTTTTAACTGATTAC-3′) were used, with a 1 µL PCR product used as a template. The cycling conditions consisted of: an initial denaturation step at 94 °C for 4 min, 25 cycles of denaturation at 94 °C for 1 min, annealing at 60 °C for 15 s, and extension at 72 °C for 15 s, followed by a final extension step at 72 °C for 10 min. The amplification products were visualized using 2% agarose gel electrophoresis.

## 3. Results

During the post mortem examination, extreme emaciation, with loss of body weight, was revealed. Various-sized granulomas and caseous plaques were observed in the lungs, air sacs, kidneys, spleen, intestinal serosa, and liver (Figure 1a,b).

The histopathological evaluation of the collected samples revealed areas of necrosis in the pulmonary parenchyma, characterized by infiltration of numerous multinucleated giant cells, epithelioid-like cells, fibrin heterophilic deposits, and hemosiderin granules. The necrotic areas were delimited from the surrounding tissue, while the bronchi spaces were blocked by inflammatory infiltrate (Figure 2a). Additionally, using PAS staining, numerous fungal hyphae with a characteristic architecture consisting of 45° dichotomous branches were observed within the necrotic and inflammatory areas of the affected lung parenchyma (Figure 2b).

In liver tissue, a diffuse inflammation, characterized by a mild congestion with a significant hepatocyte necrosis, hemosiderin granules phagocytosis, and numerous fungal hyphae, were noted (Figure 2c).

Furthermore, the cytological examination of the lung smears revealed the presence of inflammatory cells consisting of numerous neutrophiles, macrophages, and lymphocytes. Moreover, atypical cells showing multinucleated giant cells (Figure 3a) with intensely vacuolated cytoplasm and undifferentiated walls, along with few fungal hyphae, have been identified (Figure 3b).

Following the cultivation on Sabouraud dextrose agar and Mueller–Hinton agar with 5% sheep blood media, the fungal presence in the lung and liver samples was confirmed, while according to the mycological descriptions, the causative *Aspergillus* spp. was identified as *Aspergillus fumigatus* on the basis of the following: (1) cultural characteristics—the presence of dense smoky colonies, gray-green in color with a slight creamy-white reverse (Figure 4a), and (2) microscopical morphology (revealed by lactophenol cotton blue staining)—septate hyphae and columnar conidia with a diameter from 2.5 to 3 µm, basipetally constructed in chains from a single palisade-like layer of phialides that were produced directly on broadly clavate vesicles in the absence of metulae (Figure 4c). These results were confirmed by PCR and nested PCR assays, in which an amplified product of 521 bp, corresponding to genus *Aspergillus,* and a product of 310 bp, corresponding to the *Aspergillus fumigatus* ITS1 region, respectively, were amplified in both the lung and liver samples and visualized by electrophoresis (Figure 4d).

## 4. Discussion

Avian aspergillosis is described as a disease of captivity, with *Aspergillus fumigatus* and *Aspergillus flavus* mainly implicated as etiological agents, while their transmission occurs mostly via the respiratory route [10]. In our case, the infiltrate and the chronic lung parenchyma inflammation led to a progressive impaired air circulation, which eventually proved fatal. *Aspergillus* spp. is able to induce both granulomatous and diffuse lesions, consisting of some characteristic changes. Moreover, there is a high probability of finding both lesion types in the same affected organ, which may indicate that the dissemination route is not exclusively respiratory. In our case, the fresh examination of the lung granulomas using May–Grünwald–Giemsa staining demonstrated the characteristic branching structures of the *Aspergillus* spp. hyphae. Furthermore, the main lesions were identified in the lungs, where areas of necrosis, characterized by infiltration of numerous multinucleated giant cells, epithelioid-like cells, fibrin heterophilic deposits, and hemosiderin granules were noted, which may suggest that the path of conidia infection was via the respiratory route, which is stated to be the main avian infection route [3]. Beside lung infection, other organs are also reported to be affected by *Aspergillus* spp. In this regard, *Aspergillus* spp. was isolated from rib osteomyelitis in farmed ostriches, from sternum osteomyelitis in chickens, from ribs bone marrow in the African grey parrot, from kidneys in a swift parrot, from ocular lesions in domestic turkeys and Rufous owls, or from gizzard and liver lesions in the Java Finch [15,16,17]. Interestingly, in *Psittaciformes*, liver damage has not been associated with *Aspergillus* spp. infection. Accordingly, in an African grey parrot (*Psittacus erithacus*) infected with *Aspergillus fumigatus*, significant damage was found in the lungs, while no lesions were reported in the liver [4]. Moreover, a similar pattern was reported in a captive Eclestus parrot (*Eclestus roratus*) infected with *Aspergillus flavus* [10]. In our case, *Aspergillus* spp. hyphae were noted in the liver, which were accompanied by a diffuse inflammation, characterized by a mild congestion, with a significant hepatocyte necrosis and phagocytosis of the hemosiderin granules. To the authors’ best knowledge, this is the first time that *Aspergillus fumigatus* has been associated with liver inflammation, suggesting a disseminated infection. The liver involvement demonstrated in this case may be explained by the small size of the conidia, enabling them to reach the lungs and its air sacs, and frequently, the caudal–thoracic and abdominal air sacs as well. When the immune system is not able to eliminate conidia, or when their accumulation is high, the infection progresses, with subsequent dissemination to the surrounding tissues, such those of the lungs, bronchi, liver, spleen, or intestinal serosa [3,4].

The identification of the *Aspergillus* spp. is of great importance in the confirmation of the avian aspergillosis diagnostic, since confusion may occur due to the lack of differences between the lesions induced by various species of *Aspergillus* such as *Aspergillus fumigatus*, *Aspergillus flavus*, *Aspergillus niger*, *Aspergillus nidulans*, or *Aspergillus restrictus* [10,17].

According to some authors, the simultaneous isolation of *Aspergillus* spp. with the demonstration of specific lesions plays a key role in the confirmation of the diagnostic [10,18,19]. As stated by Ref. [7], “due to the ubiquity of *Aspergillus* conidia, isolation of the fungi without the presence of lesions is not of diagnostic”. In this case, the microscopical findings were further confirmed by the successful isolation of *Aspergillus fumigatus* on Sabouraud dextrose agar and Mueller–Hinton agar media and by its microscopical mycological morphology revealed by lactophenol cotton blue staining. Furthermore, the aforementioned diagnostic methods were completed by PCR using genus-specific and species-specific primers. According to Ref. [20], this approach, based on PCR detection for *Aspergillus* spp., is highly recommended for a definitive diagnosis, since this technique allows for the active demonstration of sensitivity and specificity (PCR 25%, culture 19.5%, microscopy 10%).

The histopathological findings for the macroscopical lesions, along with the fungal isolation and molecular confirmation of *Aspergillus fumigatus* by nested PCR, allowed for the diagnostic of disseminated aspergillosis.

It is important to note that the treatment of avian aspergillosis, when possible, is not always successful, for various reasons, including the following: (1) the diagnosis is confirmed in the late stages of the disease; (2) a narrow spectrum of antifungal drugs are available for avian species; (3) the available drugs may fail to penetrate the target tissues; (4) concurrent diseases and/or immunosuppression may be present [1]. Since aspergillosis occurrence in captive birds is known to depend on several factors, such as improper environmental conditions or host immunity, preventative measures play a crucial role. In this context, controlling the level of exposure and minimizing stressors are necessary. Moreover, a regular cleaning and disinfection of the nest areas, opening the canopy to increase the amount of sunlight reaching the floor of the aviaries, and ensuring proper ventilation, together with improved animal husbandry practices, can minimize stress in the environment, which in turn decreases the risk of aspergillosis [1].

## 5. Conclusions

Since birds suffering from *Aspergillus* spp. do not always show respiratory issues and considering that the clinical signs are non-specific, together, these challenges may create diagnostic difficulty for clinicians unfamiliar with the parrots’ pathology. Therefore, for a definitive diagnosis, demonstration of fungal presence by cytology or histopathology, and its identification using culturing and molecular techniques, is required.

## Figures and Tables

**Figure 1 animals-14-02282-f001:**
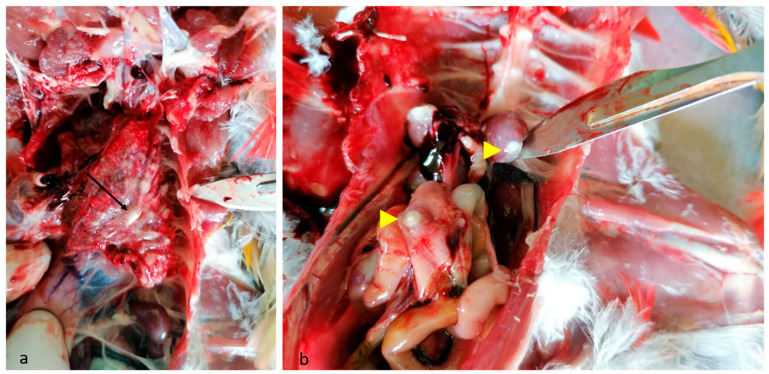
Macroscopic findings in parrot aspergillosis: (**a**) lungs showing fungal granulomas of various sizes (black arrows); (**b**) disseminated granulomas on spleen and intestinal serosa (yellow arrowheads).

**Figure 2 animals-14-02282-f002:**
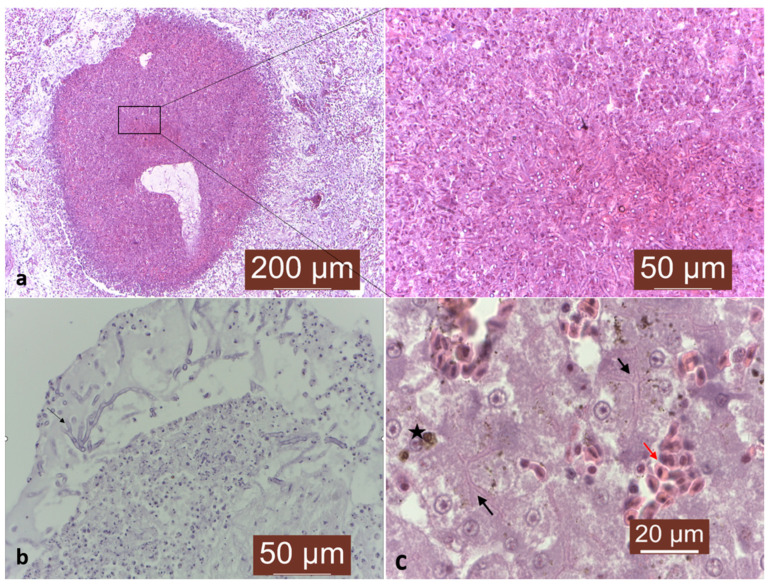
(**a**) Lung granuloma consisting of a central mass of necrotic exudate under Masson’s trichrome staining and 20× magnification. Inset shows higher magnification of granulomas, with numerous fungal hyphae under Masson’s trichrome staining and 40× magnification. (**b**) Fungal hyphae with a characteristic architecture consisting of 45° dichotomous branches in lung granuloma lesion, under PAS staining, 40× magnification. (**c**) Liver: fungal hyphae (black arrows), hemosiderin granules phagocytosis (black star), and congestion (red arrow) under Masson’s trichrome staining and 100× magnification.

**Figure 3 animals-14-02282-f003:**
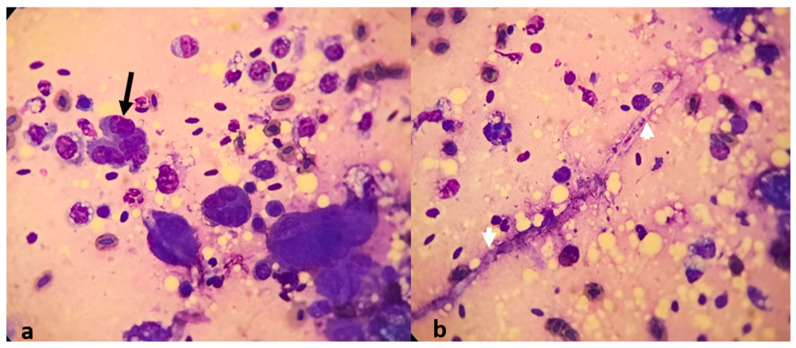
Cytology from the lung granuloma smears shows: (**a**) inflammatory cells and multinucleated giant cells (black arrow) under May–Grünwald–Giemsa staining, 40× magnification; (**b**) fungal hyphae (white arrowheads) under May-Grünwald Giemsa staining and 40× magnification.

**Figure 4 animals-14-02282-f004:**
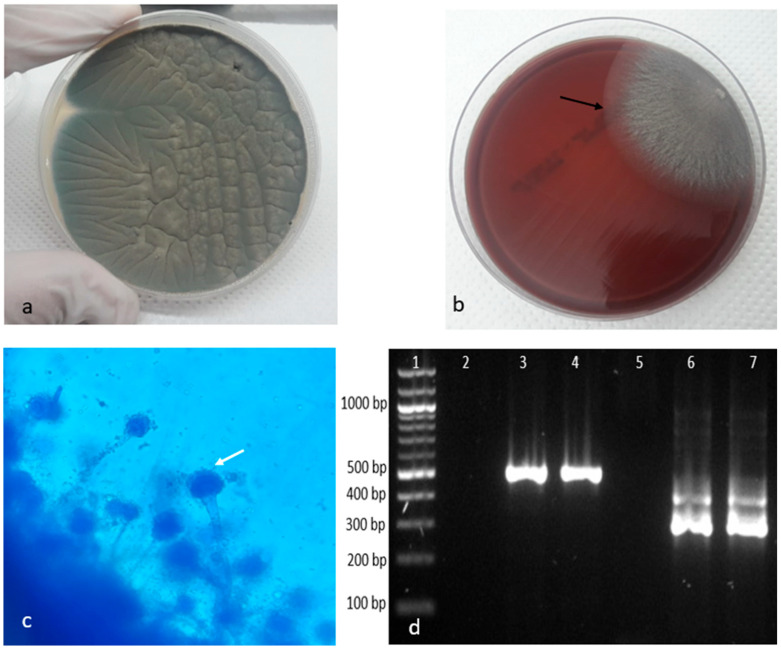
(**a**) Cultural characteristics of *Aspergillus fumigatus* colony: dense smoky and gray-green colonies on Sabouraud dextrose agar media; (**b**) cultural characteristics of *Aspergillus fumigatus* colony on Mueller–Hinton agar media showing gray-green colony (black arrow); (**c**) microscopical morphologies of *Aspergillus fumigatus* revealed by lactophenol cotton blue staining: typical columnar conidia heads (white arrow) showing characteristic form of vesicles under Lactophenol cotton blue staining, and ×20 magnification. (**d**) Molecular confirmation of genus *Aspergillus* and *Aspergillus fumigatus*: line 1: 1.2 kb molecular weight marker; lines 2 and 5 C-negative control; the product of 521 bp was amplified in line 3 (lung) and 4 (liver) samples by ASAP primers; using nested PCR, a product of 310 bp corresponding to *Aspergillus fumigatus* was obtained in the lung (line 6) and liver (line 7) samples using ASPU/AFI3 primers.

## Data Availability

The original contributions presented in the study are included in the article, further inquiries can be directed to the corresponding author.

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
