# Peer review of "Disseminated Infection with Aspergillus fumigatus in a Scarlet Macaw Parrot (Ara macao)—A Case Report"

_animals, 2024, doi:10.3390/ani14152282_

Round 1
Reviewer 1 Report
Comments and Suggestions for Authors
Dear authors
You provided a well-written manuscript describing the postmortem, microbiological, histopathologic and molecular diagnosis of systemic aspergillosis affecting respiratory and abdominal organs in a scarlet macaw (Ara macao) produced by Aspergillus fumigatus. The authors described usual gross and microscopic findings of A. fumigatus infection in a single captive macaw by using culture and molecular methodologies previously described. Despite the complete diagnosis of a catastrophic fungal disease for Psittaciformes, unfortunately, there are no novel findings from this case to contribute to the current literature of this extensively described fungal entity.
Author Response
Response to referee 1.
Indeed, few cases of aspergillosis were reported in Psittaciformes, and the main damages were found in the lungs and airsacs. As we stated in this paperwork, we detected both the fungal hyphae and the fungal DNA in liver tissue, therefore we believe the novelty is resulting from this localization. Moreover, this may help the clinicians to suspect this diseases even the respiratory signs are missing.
Reviewer 2 Report
Comments and Suggestions for Authors
The manuscript “Disseminated infection with Aspergillus fumigatus in scarlet macaw parrot (Ara macao) – a case report” reports the presence of disseminated infection with Aspergillus fumigatus in scarlet macaw which was confirmed by histopathological, microbiological and molecular methods.
Aspergillosis in avian species is a great cause of concern as it is a major cause of morbidity and mortality in birds, causing economic and ecological damages. It is unfortunate that such topics, i.e. Aspergillosis in avian species, are not often reported on in the scientific literature. However, it is an important veterinary fungal infection and can give indications for other diseases as well in line with the one health approach. In that context, the current case report is very relevant.
The authors rightly pointed out that Aspergillus fumigatus is the most prevalent species, representing up to 95% of cases, in both wild and domestic avian species. The manuscript is very straightforward in which the authors narrated the case history and confirmed Aspergillus fumigatus by histopathology, microbiological examination and PCR based assays. Overall, the writing is very good and I do not have much concern about the manuscript.
In the discussion section, authors are requested to include a para on the predisposing factors and preventive control measures.
Author Response
Reviewer #2
The manuscript “Disseminated infection with Aspergillus fumigatus in scarlet macaw parrot (Ara macao) – a case report” reports the presence of disseminated infection with Aspergillus fumigatus in scarlet macaw which was confirmed by histopathological, microbiological and molecular methods.
Aspergillosis in avian species is a great cause of concern as it is a major cause of morbidity and mortality in birds, causing economic and ecological damages. It is unfortunate that such topics, i.e. Aspergillosis in avian species, are not often reported on in the scientific literature. However, it is an important veterinary fungal infection and can give indications for other diseases as well in line with the one health approach. In that context, the current case report is very relevant.
The authors rightly pointed out that Aspergillus fumigatus is the most prevalent species, representing up to 95% of cases, in both wild and domestic avian species. The manuscript is very straightforward in which the authors narrated the case history and confirmed Aspergillus fumigatus by histopathology, microbiological examination and PCR based assays. Overall, the writing is very good and I do not have much concern about the manuscript.
Comment #1. - In the discussion section, authors are requested to include a para on the predisposing factors and preventive control measures.
Response #1 First of all, we want to thank to the referee #2 for his/her time dedicated for reviewing this case report and for his/her expertise.
Thank to the referee for this suggestion. Indeed, to improve the discussion section, we added a phrase where the predisposing factors and preventive control measures are presented. Please see lines 243-254, which are highlighted in red.
Reviewer 3 Report
Comments and Suggestions for Authors
This is a case report and the description of the pathology the isolation of the fungus appear to be appropriate. The organism was also identified using molecular techniques. This is a possible minor weakness in the study.
The English used in this presentation is commendable. However, there are some notable improvements that could be made. The authors frequently do not use the definitive article and they have not differentiated between the noun fungus and the adjective fungal.
I am not convinced that the nested PCR technique used in this study is the most precise method of determining infection with Aspergillus fumigatus. Sequencing of the larger PCR product would add additional information and confidence to the diagnosis.
The authors could have cited (Talbot et al., 2017) who noted infection in a wide range of species infected in zoos with Aspergillus fumigatus.
They used sequencing of the ITS and BenA genes and this allows the identification of the organisms with greater precision. Testing of the organism in this case using the two additional sets of primers described in (Sugita et al., 2004) would have shown that the fungus did not react with the two additional sets of primers ruling out the other two organisms as a possible aetiology.
SUGITA, C., MAKIMURA, K., UCHIDA, K., YAMAGUCHI, H. & NAGAI, A. 2004. PCR identification system for the genus Aspergillus and three major pathogenic species: Aspergillus fumigatusAspergillus flavus and Aspergillus niger. Medical Mycology, 42, 433-437.
TALBOT, J. J., THOMPSON, P., VOGELNEST, L. & BARRS, V. R. 2017. Identification of pathogenic Aspergillus isolates from captive birds in Australia. Medical Mycology, 56, 1038-1041.

The English used in this presentation is commendable. However, there are some notable improvements that could be made. The authors frequently do not use the definitive article and they have not differentiated between the noun fungus and the adjective fungal.
Title:
It may be more appropriate if this was “disseminated infection with Aspergillus fumigatus in a scarlet macaw---
Lines 19 and 27
This is fungal growth and fungal presence. The adjective is required.
Line 35
The word “symptoms” is best used for those abnormalities described by a human patient. These are clinical signs not symptoms that are referred to in this publication.
Line 38
By the presence of hyphae
Line 42
Fungal isolation
Line 45
In a scarlet macaw
Line 65
This is almost certainly haematogenous spread or transmission
Line 79
Fungal growth
Line 80
In a scarlet macaw
Line 89
Clinical signs not symptoms
Line 90
Post-mortem examination was performed and various sized granules were noted
Line 104 and 107
The names of the authors should be noted followed by the reference.
Lines 108 to 115
This description should also include the cycling parameters even if those cycling parameters were identical to those used by Sugita et al.
Line 157
Fungal presence
Line 186
By the respiratory route
Line 192
Branching structures
Line 196
By the respiratory route
Line 200
In a swift parrot
Line 202
Liver damage has not been associated
Line 204
Significant damage was found
lines 208 209
Phagocytosis of haemosiderin granules
Line 212
Which are able to reach
Line 214
The authors probably meant progressed not progress
Line 221
A key role in the confirmation of the diagnosis
Line 223
Is not diagnostic
Line 226
Morphology not morphologies
Line 227
The aforementioned diagnostic methods were
Line 228
Based on PCR detection
Line 230
A method cannot show sensitivity. This suggests the method actively participates in demonstrating sensitivity. Clearly this cannot occur the method may be more sensitive.
Line 231
Fungal isolation
Line 238
Fungal presence
Line 239
Molecular techniques
Author Response
Reviewer #3
We want to thank to the referee #3 for his/her pertinent suggestions, which will clearly improve the quality of this paper and for his/her time dedicated for reviewing this work.
- Comments and Suggestions for Authors
Comment #1. This is a case report and the description of the pathology the isolation of the fungus appears to be appropriate. The organism was also identified using molecular techniques. This is a possible minor weakness in the study.
Response #1. Thank to the observation done by the referee #3. Indeed, we used the classical protocol for aspergillosis confirmation: histopathology and cultures with species identification. Additionally, we identified the Aspergillus fumigatus by classical PCR, as suggested by other studies (Sugita et al., 2004).
Comment #2. The English used in this presentation is commendable. However, there are some notable improvements that could be made. The authors frequently do not use the definitive article and they have not differentiated between the noun fungus and the adjective fungal.
Response #2. We apologies for this error. We replaced throughout the manuscript where appropriate the fungus to fungal.
Comment #3. I am not convinced that the nested PCR technique used in this study is the most precise method of determining infection with Aspergillus fumigatus. Sequencing of the larger PCR product would add additional information and confidence to the diagnosis.
Response #3. Thank to the referee for this comment. This is a good observation, but our research funds were limited and sequencing was beyond our goal, although would bring additional information. By using the molecular method, we aimed to corelate it with the result of the culture, increasing in this way the precision of the diagnosis. Therefore, we checked the literature and we considered to be appropriate to use the protocol suggested by Sugita el al., 2004, which further was used by other researchers such Singh et al., 2016 (Raksha Singh, Gurjeet Singh and A.D. Urhekar, 2016. Detection of Aspergillus Species by Polymerase Chain Reaction. Int.J.Curr.Microbiol.App.Sci (2016) 5(10): 254-260)
Comment #4. The authors could have cited (Talbot et al., 2017) who noted infection in a wide range of species infected in zoos with Aspergillus fumigatus.
Response #4. Thank you for pointing this out. As suggested by referee #3, we already cited the work of Talbot et al., 2017, which can be found as reference number 17: Talbot, J.J.; Thompson,P.; Vogelnest, L.; Barrs, V.R. Identification of pathogenic Aspergillus isolates from captive birds in 295 Australia. Med Mycol 2018, 56(8), pp. 1038-1041. doi: 10.1093/mmy/myx137
Comment #5. They used sequencing of the ITS and BenA genes and this allows the identification of the organisms with greater precision. Testing of the organism in this case using the two additional sets of primers described in (Sugita et al., 2004) would have shown that the fungus did not react with the two additional sets of primers ruling out the other two organisms as a possible aetiology.
Response #5. Indeed, the protocol suggested by Talbot el al., 2017, targeted the ITS1 and ITS2 regions, along with BenA amplification, followed by their sequencing, thus leading to a precise identification.
Since we clearly had in our mind the implication of Aspergillus fumigatus, we ordered just the specific primers for Aspergillus fumigatus (ASPU/AFI3).
SUGITA, C., MAKIMURA, K., UCHIDA, K., YAMAGUCHI, H. & NAGAI, A. 2004. PCR identification system for the genus Aspergillus and three major pathogenic species: Aspergillus fumigatus Aspergillus flavus and Aspergillus niger. Medical Mycology, 42, 433-437.
TALBOT, J. J., THOMPSON, P., VOGELNEST, L. & BARRS, V. R. 2017. Identification of pathogenic Aspergillus isolates from captive birds in Australia. Medical Mycology, 56, 1038-1041.
- Comments on the Quality of English Language
The English used in this presentation is commendable. However, there are some notable improvements that could be made. The authors frequently do not use the definitive article and they have not differentiated between the noun fungus and the adjective fungal.
Comment #1 Title: It may be more appropriate if this was “disseminated infection with Aspergillus fumigatus in a scarlet macaw---
Response #1: We agree with this comment. Therefore, we made the change by adding ”a” in the title, which is highlighted in yellow. Please see line 2.
Comment #2 - Lines 19 and 27. This is fungal growth and fungal presence. The adjective is required.
Response #2. We apologies for this error. We replaced the fungus to fungal, which are highlighted in yellow. Please see lines 19 and 27.
Comment #3 Line 35 - The word “symptoms” is best used for those abnormalities described by a human patient. These are clinical signs not symptoms that are referred to in this publication.
Response #3. Thank you for pointing this out. We agree with this comment, therefore we changed the ”symptoms” to ”clinical signs”. Please see line 35.
Comment #4 Line 38 By the presence of hyphae
Response #4 . We apologies for this error. We added ”the”. Please see line 38.
Comment #5 Line 42 Fungal isolation.
Response #5 . We apologies for this error. We replaced ”fungus„ to ”fungal„. Please see line 42.
Comment #6. Line 45 In a scarlet macaw.
Response # 6. We agree with this comment. Therefore, we made the change by adding ”a”. Please see line 45.
Comment #7. Line 65. This is almost certainly haematogenous spread or transmission.
Response #7. We agree with this comment. Accordingly, we made the change as highlighted in yellow. Please see lines 65-66.
Comment #8 Line 79 Fungal growth.
Response #8 . We apologies for this error. We replaced ”fungus „ to ”fungal„. Please see line 79.
Comment #9 Line 80 In a scarlet macaw.
Response # 9. We agree with this comment. Therefore, we made the change by adding ”a”. Please see line 80.
Comment #10 Line 89 Clinical signs not symptoms.
Response #10. Thank you for pointing this out. We agree with this comment, therefore we changed the ”symptoms” to ”clinical signs”. Please see line 89.
Comment #11 Line 90 Post-mortem examination was performed and various sized granules were noted
Response #11. Thank to the referee for this suggestion. We made the change accordingly ” Post-mortem examination was performed and various sized granules were noted”. Please see line 90.
Comment #12 Line 104 and 107 The names of the authors should be noted followed by the reference.
Response #12. Thank you for pointing this out. We agree with this comment, therefore we added the name of the authors Thom and Raper (1945) and Sugita et al. (2004). Please see lines 104-105 and 108.
Comment #13 Lines 108 to 115. This description should also include the cycling parameters even if those cycling parameters were identical to those used by Sugita et al.
Response #13. Thanks to the referee for this We agree with this comment. Accordingly, we made the change as highlighted in yellow. We added the phrase describing the PCR conditions for ASAP primers: The cycling conditions were consisting in: an initial denaturation step at 940 C for 4 minutes, 30 cycles of denaturation at 940 C for 1 minute, annealing at 550 C for 2 minutes, and extension at 720 C for 90 seconds, followed by a final extension step at 720 C for 10 minutes. Please see lines 112-115.
For ASPU/AFI3 primers the next phrase was introduced: The cycling conditions were consisting in: an initial denaturation step at 940 C for 4 minutes, 25 cycles of denaturation at 940 C for 1 minute, annealing at 600 C for 15 seconds, and extension at 720 C for 15 seconds, followed by a final extension step at 720 C for 10 minutes. Please see lines 118-121.
Comment #14 Line 157 Fungal presence
Response #14. We apologies for this error. We replaced ”fungus „ to ”fungal„. Please see line 164.
Comment #15 Line 186 By the respiratory route
Response # 15. We agree with this comment. Therefore, we made the change by adding ”the”. Please see line 193.
Comment #16Line 192 Branching structures
Response # 16 We agree with this comment. Therefore, we made the change by adding ”branching structures”. Please see line 199-200.
Comment #17 Line 196 By the respiratory route
Response # 17. We agree with this comment. Therefore, we made the change by adding ”the”. Please see line 203.
Comment #18 Line 200 In a swift parrot
Response # 18. We agree with this comment. Therefore, we made the change by adding ”a”. Please see line 207
Comment #19 Line 202 - Liver damage has not been associated
Response # 19. We agree with this comment. Therefore, we made the corrections as suggested ” liver damage has not been”. Please see line 209
Comment #20 Line 204 Significant damage was found
Response # 20. We agree with this comment. Therefore, we made the corrections as suggested ” significant damage was found”. Please see line 211.
Comment #21 lines 208 209 Phagocytosis of haemosiderin granules
Response #21. We agree with this comment. Therefore, we made the corrections as suggested ”and phagocytosis of hemosiderin granules”. Please see line 215-216.
Comment #22 Line 212 Which are able to reach
Response #22. We agree with this comment. Therefore, we made the corrections as suggested ” which are able”. Please see line 219.
Comment #23 Line 214 The authors probably meant progressed not progress
Response #23. We apologies for this error. We replaced ”infection progress „ to ”infection progressed„. Please see line 221.
Comment #24 Line 221 A key role in the confirmation of the diagnosis
Response #24. We apologies for this error. We replaced ”a key role in diagnosis confirmation„ to ” a key role in the confirmation of the diagnosis „. Please see line 229.
Comment #25 Line 223 Is not diagnostic
Response #25. We agree with this comment. Therefore, we made the corrections as suggested ” not of diagnostic”. Please see line 231.
Comment #26 Line 226 Morphology not morphologies
Response #26. We apologies for this error. We replaced ”morphologies„ to ” morphology „. Please see line 234.
Comment #27 Line 227 The aforementioned diagnostic methods were
Response #27. We agree with this comment. Therefore, we made the corrections as suggested ” aforementioned diagnostic methods”. Please see line 235.
Comment #28 Line 228 Based on PCR detection
Response #28. We apologies for this error. We replaced ”in„ to” on „. Please see line 236.
Comment #29 Line 230 - A method cannot show sensitivity. This suggests the method actively participates in demonstrating sensitivity. Clearly this cannot occur the method may be more sensitive.
Response #29. Thanks to the referee suggestions. For a better understanding of this sentence, we reformulated it accordingly as follow ” According to [20] this approach, based on PCR detection for Aspergillus spp. is highly recommended for a definitive diagnosis, since this technique actively participates in demonstrating sensitivity and specificity (PCR 25%, culture 19,5%, microscopy 10%).” Please see lines 237-238.
Comment #30 Line 231 Fungal isolation
Response #30. We apologies for this error. We replaced ”fungus „ o”fungal„. Please see line 240.
Comment #31 Line 238 Fungal presence
Response #31. We apologies for this error. We replaced ”fungus „ to ”fungal„. Please see line 260.
Comment #32 Line 239 Molecular techniques
Response #32. We apologies for this error. We replaced ” Molecular technique „to” Molecular techniques„. Please see line 261.
Round 2
Reviewer 1 Report
Comments and Suggestions for Authors
Based on my previous review I suggested rejection on this manuscript, particularly due to the lack of original findings. Please select another reviewer to continue with the revision because I disagree to continue with this revision.